# A Study on the Development Trends of the Energy System with Blockchain Technology Using Patent Analysis

**Lin-Yun Huang [1], Jian-Feng Cai [2], Tien-Chen Lee [3] and Min-Hang Weng [1,*]** 

[1] School of Information Engineering, Putian University, Putian 351100, Fujian, China; huanglinyun@ptu.edu.cn
[2] Office of Digital Putian Leading Group, Putian 351100, Fujian, China; caijiangfeng001@gmail.com
[3] Fuzhou University of International Studies and Trade, Changle 350202, Fujian, China; ltc@fzfu.edu.cn
[*] Correspondence: hcwweng@gmail.com

**Abstract:** Recently, the application of blockchain to the setting, management, and trading of the energy system has formed an innovative technology and has attracted a lot of attention from industry, academia, and research. In this study, we use patent analysis technology to explore the development trends of the energy system with blockchain technology. During the patent analysis process, this study makes corresponding analysis charts, such as patent application numbers over time, patent application numbers for main leading countries, applicants, patent citations, international patent classification (IPC), and life cycle. Relative research and design (R&D) capability of the top ten applicants is estimated and the cluster map of the technology is obtained. The technical features of the top five IPC patent applications are related to the cluster map to show the development of energy blockchain technology. Through this paper, first, the basics of the blockchain and patent analysis are illustrated and, moreover, the reason why and how blockchain technology can be combined with the energy system is also briefly described and analyzed. The results of the patent analysis of energy blockchain technology indicate that the United States leads the way, accounting for more than half of the global total. It is also interesting to note that the participants are not from traditional specific fields, but included electric power manufacturers, computer software companies, e-commerce companies, and even many new companies devoted to blockchain technology. Walmart Apollo, LLC and International Business Machines Corporation (IBM) have the highest number of patent applications. However, Walmart Apollo, LLC ranks first with a greater number of inventors of 36, an activity year of 2 years, and a relative R&D capability of 100%. IBM ranks second with an activity year of 3 years and a research and development capability of 91%. Among various applicants, IBM and LO3 energy started earlier in this field, and their patent output is also more prominent. The IPC is mainly concentrated in G06Q 50/06, which belongs to the technical field of the setting and management of the energy system including electricity, gas, or water supply. Currently, most projects are in the early development stages, and research on key areas is still ongoing to improve the required scalability, decentralization, and security. Thus, energy blockchain technology is still in the growth period, and there is still considerable room for development of the patent in the later period. Moreover, it is suggested that the novel communication module such as the combination of the consortium blockchain and the private blockchain cold also provide their own advantages to achieve the purpose of improving system performance and efficiency.

**Keywords:** blockchain; energy system; patent search; patent analysis

---

## 1. Introduction

In recent years, innovative technologies have revolutionized the financial industry with the development and application of next-generation technologies such as mobile internet, big data, cloud computing, blockchain, artificial intelligence (AI), and the Internet of Things (IoT), followed by logistics, medical care, energy, and other fields that have also begun to be gradually applied [1,2]. Blockchain technology, especially, is the disruptive innovation on the financial economy, and has been addressed as a next generation alternative to replace the internet as a new network architecture. The emergence of blockchain technology has fundamentally overturned the inherent logic, operating mode, and business scope of traditional finance. Blockchain is a kind of underlying technology of Bitcoin, which is actually a decentralized trust mechanism. Through the blockchain, both parties in the transaction can carry out economic activities without the use of third-party credit intermediaries, thereby reducing the cost of globally transferred assets. Currently, blockchain is applicable to the financial field, and also applicable to the energy field [3].

In other industries, renewable energy is a key area supported by green finance, and its application in finance is also increasing [4–6]. The development of renewable energy has promoted enterprises' investment in low-carbon economies, that is, through relevant policies, companies that focus on sustainable development can obtain financing and encourage financing to enter new areas [4]. New research questions are formulated about how finance affects the directionality of innovation, and the implications for the policies of renewable energy [4]. At present, renewable energy technology is still rapidly advancing and its costs continue to decrease. It is entering a new development cycle, which also brings opportunities to related industries. The universalization and miniaturization of renewable energy projects makes investment space huge. Both enterprises and individuals can invest in them and share the benefits of renewable energy [6].

The traditional electric power trading model relies on third-party institutions, and the trading process is complicated, low in efficiency, and lengthy [7]. Moreover, the trading process causes large losses, high transaction costs, and low security in the process of electric power transportation. Since blockchain technology has the characteristics of decentralization, transparency, fairness, and openness, these characteristics are similar with the concept of the Energy Internet. Applying blockchain technology to the Energy Internet has a significant impact on the construction and development of the Energy Internet [8]. Energy system combined with blockchain technology uses high-speed computing algorithms and smart meters to integrate electricity in the distributed grid, obtain real-time information on both side between the energy generators and energy users, and allow electricity to enter the power-sharing economy through the smart contracts in the blockchain platform [9]. In this way, ordinary households can get clean and renewable energy in a free, equal and real-time electricity market mechanism. The core technology of the service model is to provide a decentralized point-to-point (P2P) power usage calculation, pricing, and fee payment service through a blockchain platform [10,11]. Each node has complete data, rights and obligations, and manages the entire ecological operation of the energy system and, then, achieves the goal of "users are the producers and also the consumers" [12,13].

In recent years, domestic and foreign countries have invested in related patent applications during the take-off phase of the energy system with blockchain technology. Since the patent quality and layout strategy of the blockchain in the energy field affect subsequent widespread applications, the information provided in patent literature is a valuable resource for discovering the development trend of specific technological area. Although patent information is noteworthy to evaluate the development trends, there are still few studies using the most recent patent documents to discuss the development trends of the energy system combined with blockchain technology. Due to the rapid development of the energy system combined with blockchain technology and the high potential of the industrial applications in this field, it is necessary to realize the following questions:

1. What are the countries/companies/applicants in the existing patent application layout of the energy system combined with blockchain technology?

2.　　What is the technical distribution or cluster of the application in the existing patent application of the energy system combined with blockchain technology?

3.　　What are the development directions that have appeared in the following patent application of the energy system combined with blockchain technology?

Recently, most of the related literatures have focused on a general review of all research activities, application methods, and potential of blockchain technology in the energy system.

Patent analysis is known as an effective method with patent visualization that helps participants know useful legal, economic, and technical information from patent documents. Therefore, the purpose of this study is to analyze the development trends of the energy system combined with blockchain technology through patent analysis. The main difference between the approach of this study and previous studies is that we use patent analysis with technology cluster map in order to classify the development trends. The study could help practitioners in this area to achieve the following tasks: Searching the leading country/region, classifying key competitors, as well as identifying technology cluster. Through this article, the current status and direction of the development of the energy system with blockchain technology are described and the leading countries and enterprises in the current field of energy blockchain are disclosed. Moreover, the overview of the main patent layouts of key technologies in the energy blockchain are provided as a basis for future research.

The organization of this paper is described as follows: The first part introduces the research background and motivation of this study concerning the energy system with blockchain technology; the second part introduces the literature review about the relevant technologies to be mentioned in the research, including the significance of blockchain, blockchain based energy system, patents, and patent analysis; the third part introduces the research architecture, the research contents, and the patent search scheme; the fourth part shows the analyzed results; and the final part summaries the findings of the entire research and recommendations of the patent layout strategy.

## 2. Literature Review

### 2.1. Blockchain

Recently, blockchain technology has become a new epiphany, as a disruptive innovation on the financial economy. Blockchain technology is widely used, from national land planning to personal land registration, tracking of valuables such as diamonds, verification statements for financial statements, auto insurance, medical care, energy trade, etc.

Blockchain technology is a technology solution that does not rely on third parties and uses its own distributed nodes to store, verify, transfer, and communicate network data. Therefore, from the perspective of financial accounting, blockchain technology is regard as a distributed and open decentralized large-scale bookkeeping network. Anyone can use the same technical standard to add their own information at any time. The blockchain continues to meet various recording, verification, and write-off needs. In short, blockchain technology refers to a way for people to participate in bookkeeping [14–16].

Figure 1 shows the change diagram from Blockchain 1.0 to Blockchain 4.0 [1,8]. With respect to the Blockchain 1.0 concept, the distributed ledger technology (DLT) provides the foundation for the blockchain platform in digital currency applications, especially Bitcoin, which is used as "Internet Cash". Such a digital payment system is seen as the promoter of the "currency internet". With respect to the Blockchain 2.0 concept, the new impression proposed is a smart contract, an automated computer program that executes automatically. An important advantage of Blockchain 2.0 is that it is impossible to tamper with or crack smart contracts. As a result, smart contracts reduce the costs of verification, enforcement, arbitration, and fraud prevention, and thus allow transparent contract definitions to overcome the moral hazard issues. The most prominent of the example in this field is the execution of smart contracts in the Ethereum blockchain. With respect to the Blockchain 3.0 concept, the decentralized applications (DApp) avoids the centralized infrastructure and uses high speed and

expandable decentralized storage and decentralized communication. The back-end code of most Dapps runs on the decentralized peer-to-peer network, that is, the blockchain network. In contrast, the back-end code for traditional applications runs on a centralized server. Thus, Dapp indicates the combination of front-end code and the contract, running on Ethereum. With respect to the Blockchain 4.0 concept, the described solutions and methods make blockchain technology available to business needs, especially the requirements of industry, namely, blockchain is expected to be available in Industry (4.0) with new ecosystem, building on the foundation of previous versions [1,16].

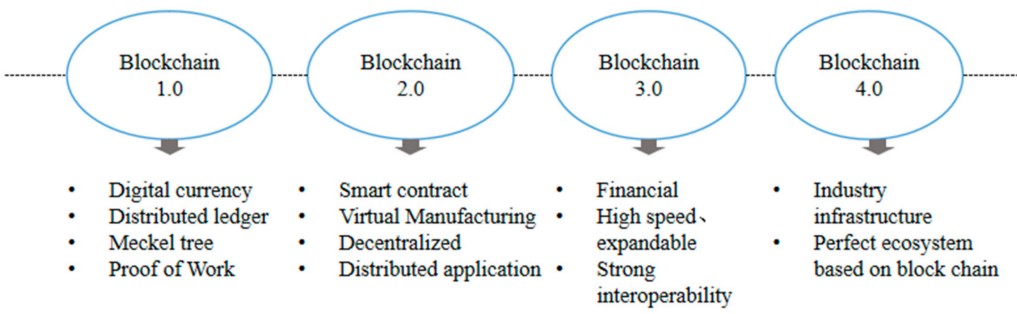

**Figure 1.** The change diagram of Blockchain 1.0 to Blockchain 4.0.

## 2.2. Energy System with Blockchain Technology

Since blockchain technology has the characteristics of decentralization, transparency, fairness, and openness, these characteristics are similar to the concept of the Energy Internet. Applying blockchain technology to the Energy Internet has a great impact on the construction and development of the Energy Internet [8,17]. In addition, the highly decentralized, highly secure method for recording transactions has proven to be faster and more secure than the traditional central architecture used today that requires coordination and approval of transactions. Blockchain can add renewable energy and other distributed energy sources to the power system, improving the visibility and control of distributed energy sources to meet the increasingly complex power grid operation needs [18,19].

Energy blockchain is a term derived from the maturity of the application of blockchain in the energy field. At present, academia has not formed a unified definition of energy blockchain. However, generally speaking, energy blockchain technology refers to the application of blockchain technology in the energy system, including system setting, management, and trading [20]. Specifically, energy includes power, oil, natural gas, cooling and heating, and other renewable energy subsystem. Thus, in the following, the energy system combined with blockchain technology is referred to simply as the energy blockchain [21].

Energy blockchain technology is a revolutionary decentralized energy interconnected data structure, in which nodes of the energy blockchain (1) uses orderly linked encrypted blocks to verify and store related energy transaction data information, (2) uses consensus mechanisms to make the distributed decisions and maintain the network-wide data, (3) uses smart contracts to automatically complete the transfer of the data information, and (4) processes the mutual verification or execution [22]. In addition, the advantage of the blockchain-based energy trading model lies in the use of P2P direct energy trading, which reduces transaction costs and reduces power consumption. The non-tampering feature greatly improves security, making energy transactions safe, transparent, and convenient [23].

Figure 2 shows a blockchain-based energy management and trading model [11,23,24]. The main components of the energy blockchain technology are the energy blockchain network (EBN), the energy supply index blockchain (ESIB), and the energy trading blockchain (ETB), which are linked with several nodes. There are three types of energy nodes which include seller nodes, buyer nodes, and idle nodes. Depending on the energy needs and status, multiple roles are played in the energy blockchain. The blockchain-based energy transaction model is a distributed energy transaction based on P2P. The transaction subject is initiated by the seller (power plant, power company, distributed energy

producer, etc.) with the energy supply status and is also initiated by the buyer with the purchase demand on the transaction layer of the blockchain. Then, the transaction subject negotiates, reaches the transaction intention, and thus forms a smart contract. The smart contract is then broadcast to each node of the energy blockchain through P2P. When the execution conditions of the smart contract are met, the contract is automatically executed to improve transaction efficiency [24].

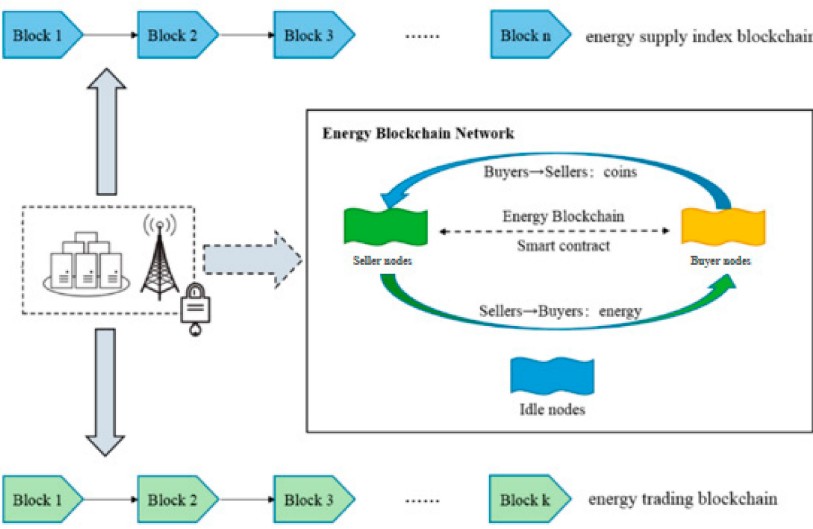

**Figure 2.** A blockchain-based energy management and trading model.

Energy blockchain technology combines three areas of domain knowledge: energy, blockchain, and business models. Namely, energy blockchain technology is an innovative peer-to-peer business model that combines energy systems with blockchain platforms at the same time. In this field, patent directions can be researched and developed from many aspects, including the setting, storage, control, transmission, and management of various energy systems; communication module, data storage and computing method and device of the network; and the incentive scheme and information security of the blockchain [22–24].

## 2.3. Patent Analysis

A patent is a right granted to the research and design (R&D) person and a channel for the development of science and technology. The purpose of setting up patents is to allow inventors to prohibit others from making, using, or selling their inventions within a certain period of time, and therefore to protect the rights of the inventors [25]. A group of patents in a particular technology represents the scientific and technological knowledge accumulated in that technology. An increase in the number of patents for a certain technology effectivley reflects the development of the technology. Therefore, patent data is widely used in technology assessment and forecasting [26]. In today's corporate competition, intellectual property is one of the most powerful tools for the world's major manufacturers to contain competitors. The most common of these is the lawsuit for patent infringement. At the beginning of production, if a manufacturer fails to carry out a detailed patent analysis of the relevant technology, it will face a painful price in the face of foreign manufacturers' recovery of royalties [27]. For the industry, the more detailed the patent information, the easier and more accurate it is for manufacturers to draw up the company's business strategy, R&D strategy, and patent strategy. Therefore, how to conduct a thorough patent analysis and planning at the beginning of research and development are areas that business leaders must pay close attention to [28,29].

Patent analysis uses patent documents to assist the researchers as a reference for the research and the development, as well as the investment management. Patent analysis is a method of systematically organizing patent information. By using map-based visualization effects, complex patent-related

information is represented on the map in a two-dimensional manner, thus enabling readers to understand related events [30,31]. Patent analysis is an effective tool for a technology research and development plan, as well as the management of intellectual property rights, and can also be used as a basis for technical competition analysis and technology trend analysis [32]. For example, a patent analysis of the graphene industry is conducted to obtain a sustainable competitive advantage from the patent information [33]. Technology competition of graphene biomedical technology has been analyzed based on patent analysis. Patent analysis with text mining was used to forecast emerging technologies in wireless power transfer [34].

Typically, the patent number, inventive countries, inventive companies, inventors, citation rates, life cycle, etc. can be used to explore the technology development of the selected subject, the technology maturity of the industry, the degree of the industrialization, the resistance encountered after entering the subject industry, the main competition opponents, etc. [35,36]. From the patent analysis, the following can be found: (1) the trend of technological development, (2) the dynamics of competitors, (3) the evaluation of investment opportunities, and (4) the basis for research and development management. In short, patent analysis is a systematic analysis process that transforms patent information into patent intelligence and is a powerful tool for research planning to resolve the crisis of infringement and enhance competitiveness [37].

In this paper, through a comprehensive patent-related chart analysis, the current development trend of energy blockchain technology is revealed.

## 3. Research Architecture and Methods

### 3.1. Research Methodology

Figure 3 shows the main steps of the methodology of patent analysis for the energy blockchain technology. The methodology is classified into four major steps as following: (1) identifying research topics, (2) determining the patent search strategy, (3) screening of patent, and (4) drawing the patent analysis charts.

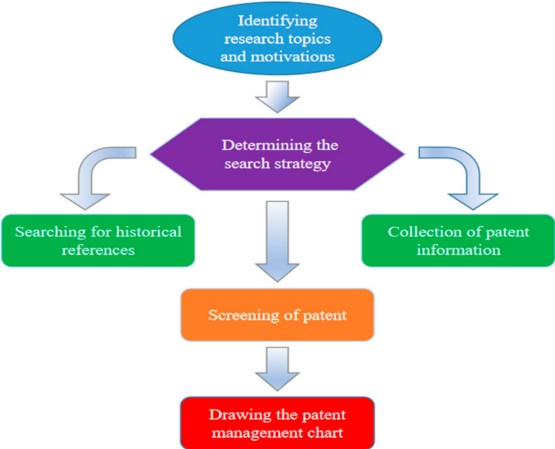

**Figure 3.** The main steps of the methodology of patent analysis for the energy blockchain technology.

In the first step, identifying research topics is used to determine the search scope of the energy blockchain technology. In this part, the related data are collected first to understand the current development status of energy blockchain technology based on the scope of the topic, the principles, and technologies of the topic. On the basis of the collected information, the technological keywords of the energy blockchain technology are further summarized and classified [36].

The second step of determining the patent search strategy is to formulate a search strategy based on the information summarized and classified in the first part, determine the keywords and items to be searched, perform preliminary reading on the retrieved patent data and delete some patents that do

not fit the subject, and also revise the search keywords and items based on the screening results [37–39]. Then, based on the revised keywords for search in the patent database, various patent analysis charts are created.

In the third step, the patents are screened to find the technology items in the retrieved patent data and then obtain a technology cluster map by using text mining to relate with the most used international patent classification (IPC) to obtain the development trend [40–42].

In the fourth step, the patent analysis chart is drawn to organize the patent analysis chart from the results of the patent search, summarize the collected theme area, and understand the patent layout of the leading unit in accordance with the patent analysis chart. On the basis of the third step of screening of patent, the patent layout strategy is also evaluated to give suggestions on the patent application of the related technological hotspots [43].

### 3.2. Research Contents

The research contents of this study are to analyze the patents related to energy blockchain technology and to evaluate a certain development direction for the energy blockchain in the future. The information obtained through the patent analysis is sorted into an important reference basis for decision-making units to formulate R&D and business strategies. In the analysis section, the research contents of patent analysis for the energy blockchain technology are presented as the following [39–42]:

1. The analysis of patent number over the years is completed by analyzing the patent number in each application year and publication year; this analysis observes the patent development status of energy blockchain technology invested by the participants in this field over time. The patent application date refers to the date when the patent was filed, and the patent publication date refers to the date on which the patent application is early published.
2. The analysis of country is used to explore the technical development of energy blockchain technology over time in various countries, and mainly understand the information and markets of leading countries of the energy blockchain technology.
3. The analysis of patentees involves analyzing various competition indicators for the specific competitors. The analysis includes statistics on the number of patents of important companies, which deeply understand the development trends of competitors' R&D strategies and are used as a reference for the company's development.
4. The analysis of IPC classification is necessary because, typically, it is not easy for patent analysts to evaluate and screen additional secondary data in the found patents. Therefore, international patent classification (IPC) helps the patent analysts through a more accurate search and screening, and also classifies the technology attributes of the found patents.
5. The analysis of life cycle is conducted to show which period the research field is in base on the number of patent applications and the number of applicants, including (1) the introduction period, (2) the growth period, (3) the mature period, or (4) the decline period.
6. The cluster map is obtained using text mining on the abstract and the summary of the invention of the found patents, thereby the retrieved patent applications are divided into several technology clusters on a map.

### 3.3. Patent Search Strategy

It is well known that in patent analysis the step of collecting accurate patent data plays a crucial role in exploring the right patent information. The search scope set by this research is from the emergence of the energy system with blockchain technology from 2008 to the present. This study uses IPTECH software provided by the innoVue company of Taiwan to search the related application and published patents regarding energy blockchain technology based on a global database.

On the basis of the collected information, the searching keywords of the energy blockchain are set mainly as "blockchain", "energy*", and "electr*".

　　　The first step is a preliminary search in a large range by searching the searching keywords in many search items including "Title", "Abstract" and "Claim".

　　　The second step is to screen for "IPC", based in the searching result of the first step. After screening, the G06Q 50/06 of the main IPC for the most searched result is selected.

　　　The third step is to rationalize the search formula by selecting the following four aspects: blockchain, energy, electric, and IPC.

　　　Table 1 shows the search formula and related patent numbers in the field of energy blockchain. From Table 1, it is found that as the searching keywords of the energy blockchain were combined with the main IPC, the number of the searching results tended to a stable value, which indicated that the search scheme was reliable and stable.

**Table 1.** Search formula and related patent numbers in the field of energy blockchain.

| Search Formula | Related Patent Numbers Approved/Public |
|---|---|
| ("blockchain " < IN > TACD) | 1468/10746 |
| ("blockchain" < IN > TACD) AND ("Energ*" < IN > ABST) | 13/143 |
| ("blockchain" < IN > ABST) AND ("Energ*" < IN > CLMS) | 0/43 |
| ("blockchain" < IN > ABST) AND ("electr*"< IN > CLMS) | 0/19 |
| ("blockchain" < IN > TACD) AND ("energy*" < IN > CLMS) OR ("blockchain" < IN > TACD) AND ("electr*"< IN > CLMS) | 16/256 |
| ("blockchain" < IN > TACD) AND ("G06Q 50*" < IN > IC) | 9/74 |
| ("blockchain" < IN > TACD) AND ("G06Q 50*" < IN > IC) OR ("blockchain" < IN > TACD) AND ("energy*" < IN > ABST) OR ("blockchain" < IN > TACD) AND ("electr*"< IN > ABST) | 17/171 |
| ("blockchain" < IN > TACD) AND ("G06Q 50*" < IN > IC) OR ("blockchain" < IN > TACD) AND ("energy*" <IN > CLIMS) OR ("blockchain" < IN > TACD) AND ("electr*"< IN > CLIMS) | 22/288 |
| ((("blockchain" < IN > TACD) AND ("G06Q 50*" < IN > IC) OR (("blockchain" < IN > TACD) AND ("energy*" < IN > ABST) OR (("blockchain" < IN > TACD) AND ("electr*"< IN > ABST)) OR (("blockchain" < IN > TACD) AND ("G06Q 50*" <IN > IC) OR (("blockchain" < IN > TACD) AND ("energy*" < IN > CLIMS) OR ("blockchain" < IN > TACD) AND ("electr*"< IN > CLIMS)) | 24/330 |

Note: ABST = Abstract; CLMS = claim(s) ; IC = IPC/LOC; TACD = Title/Abstract/Claim/Description.

　　　Finally, the searching results are up to 5 December 2019, and the number of patents found related to energy blockchain technology is 330. It is worth mentioning that the search results have certain limitations due to the limitation of time and geographical patent terms. Namely, the number of patents in 2019 could be incomplete due to the 18 month lag period of patents. With a preliminary interpretation of the abstract and the summary of the invention of the found patents by manual means, a small number of less relevant patents were eliminated. After deleting some patents that did not fit the subject, the number of found patents to be analyzed is 319. Among them, 24 patents were approved and 295 were public and disclosed. Since only 24 patents were approved, therefore, this study performed data analysis based on 319 patents which were now disclosed due to early publication. It is noted that since a patent application does not necessarily result in an approved patent through the examination, however, because the number of the approved patents is quite small, we analyze the number of patent applications by using the method of analyzing the number of the approved patents in the past. It could be an error in the evaluation of the relative R&D capability, but it has little effect on the analysis of development trends.

　　　After determining 319 patents as the final selection, the title, patent number, abstract, inventor, inventor's country, patentee, patentee country, and IPC, foreign priority, main examiner, etc, of the

found energy blockchain is obtained. On the basis of the patent data of the found patents, different patent analysis charts are made to facilitate the formation of the energy blockchain technology.

## 4. Results and Discussion

### 4.1. Analysis of the Number of Patents over the Years

According to the number of patents applied in the past years, the relevant industry information in each age and the resource investment trends of relevant competing companies can be observed. Figure 4 shows the trend of the number of patents of energy blockchain technology over the years. It is observed that from 2015 to 2016, the number of the related filed patents did not exceed 30 during that year, and there was no large number of public patent outputs. Thus, it was identified as the technology germination period of energy blockchain technology. Filed patent applications have increased to 53 since 2017, which indicates that energy blockchain technology has gradually begun to develop. The number of filed patent applications from 2017 to 2018 has increased significantly, which indicates this technology has begun to gain the attention of enterprises or research units. Especially in 2018, the number of filed patent applications in the field of energy blockchain technology reached 173. However, the number of public patent applications has increased significantly from 2017 to 2019. It is noted that the date of the public patent application is typically eighteen months later than the date of the filed patent application. According to the data, in the past two years, the technology of energy blockchain technology has gradually grown and has gradually begun to mature.

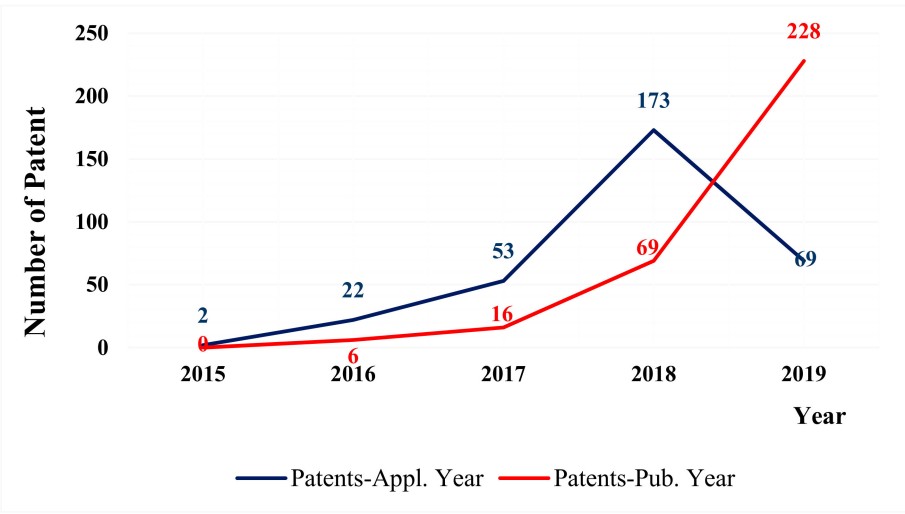

**Figure 4.** The trend of the number of patents in energy blockchain technology over the years.

### 4.2. Analysis of Country

According to the patent application status of each country in energy blockchain technology, the development status of energy blockchain technology for each country is observed.

Figure 5 shows the share of the patent applications in energy blockchain technology for the countries. From Figure 5, we found that United States (USA) has the highest share of energy blockchain technology patent applications with up to 59%, followed by Germany (DE) with 27.9%, and China (CN) ranks third with 6%, Great Britain (GB) and Israel (IT) have 5% and 2%, respectively, and the remaining 19% come from other countries. According to the results of shares by country, we see that in terms of the patent layout of energy blockchain technology, the relevant implementations in the United States, Germany, and China are the most complete.

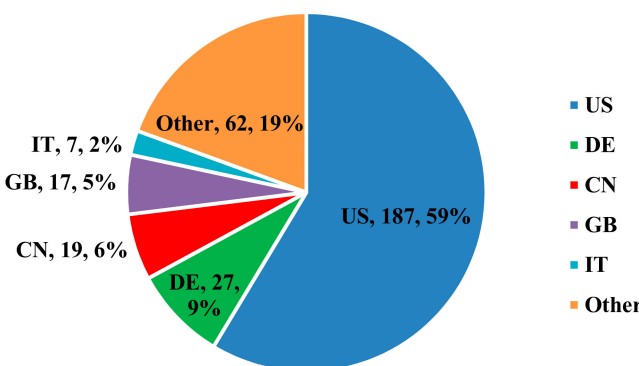

**Figure 5.** The share of patent applications in energy blockchain technology by country.

*4.3. Analysis of Patentees and Applicant*

The patentee refers to the applicant who filed a patent in his own name for his own rights, and the patentee is the owner of the patent. When a patent application is approved and announced, the applicant obtains the patent right and becomes the patentee. The inventor refers to the substantial contributor to the patented invention.

In this analysis, the applicant is the object of analysis. The purpose is to find the investment company and patentee for energy blockchain technology, which is more likely to be the object of future patent litigation. By combining detailed data values such as the number of patents, the number of citations by others, the number of self-citations, the inventor number, the country number, the average age of patents, the activity year, and the R&D capability of the competing units in specific technology fields can be fully revealed. The state of investment helps companies to develop important strategies such as technological development, production, or sales. Table 2 shows the meanings in specific fields comprising the number of patents, the number of citations by others, the number of self-citations, the inventor number, the country number, the average age of patents, the activity year, and the R&D capability. In this study, the R&D capability is estimated by using the addition of different weights of the number of patents, the number of citations by others, the number of self-citations, the inventor number, the country number, the average age of patents and the activity year. It is noted that the R&D capability is only provided as a reference since the found patent applications are public and may not be approved to be the allowed patent.

**Table 2.** The meanings in specific technical fields.

| | |
|---|---|
| Number of patent application (*pc*) | The number of patents in the company's project. The company has a large number of patents, indicating that the company has better research and development strength in the industry technology. |
| Number of citations (*oc*) | The total number of times a patent owned by a given company has been cited. The more often cited, the greater the value of the patent, which may be the foundation or core patent in the technical field. |
| Number of self-citations (*sc*) | Number of self-citations refers to the number of patents of the company in the company's citation project. The greater the number of self-citations, it means that the company focuses on self-development, but has limited technical interaction with the outside world. |
| Average patent age (*py*) | The average patent age is the sum of the patent ages divided by the number of patents. The younger the patent age, the more the company enjoys a longer technological monopoly advantage in the field, and vice versa. |
| Activities period (*ay*) | It means that the patentee has a long period of patent output activities in this technical field, and this can be used to evaluate the years that the patentee has invested in research and development in this technical field. |
| Relative R&D capability (*Xi*). | Relative R&D capability in this study is estimated using the following equation. $X_i = (5pc + 2oc + 1sc + 1ic - 1py + 0ay)$ (ic: invention number weighting parameter) M = MAX(*Xi*) Relative R&D capabilities = Xi / M |

Table 3 shows the analysis of the top ten applicants for the patent R&D intensity of energy blockchain technology, including Walmart Apollo, LLC; International Business Machines Corporation

(IBM); General Electric Company; Strong Force IoT Portfolio 2016, LLC; Ethicon LLC; Siemens Aktiengesellschaft; Hepu Technology Development (Beijing) Co. Ltd.; JPMorgan Chase Bank, N.A.; Strong Force TX Portfolio 2018, LLC; and LO3 Energy INC. This result is very interesting, showing that the participants are not from traditional specific fields, but include electric power manufacturers, computer software companies, e-commerce companies, and even many new developed companies devoted to blockchain technology. This result shows that energy blockchain technology is an extremely innovative field, and companies in different fields have the opportunity to invest in this new field of innovation and technology, as long as they can find new niche points.

**Table 3.** Analysis of the top ten applicants for the patent Research & Developing intensity of energy blockchain technology.

| Applicant | Number of Patent Application | Others Citings | Self Citings | Inventor Number | Country Number | Patent Age | Activity Year | Relative R&D Capability |
|---|---|---|---|---|---|---|---|---|
| Walmart Apollo, LLC | 9 | 0 | 0 | 36 | 1 | 0 | 2 | 100% |
| International Business Machines Corporation | 9 | 0 | 0 | 31 | 1 | 2 | 3 | 91% |
| General Electric Company | 7 | 0 | 0 | 18 | 1 | 1 | 3 | 64% |
| Strong Force IoT Portfolio 2016, LLC | 7 | 0 | 0 | 11 | 1 | 1 | 2 | 56% |
| Ethicon LLC | 7 | 0 | 0 | 9 | 1 | 1 | 1 | 53% |
| Siemens Aktiengesellschaft | 6 | 0 | 0 | 11 | 1 | 1 | 2 | 49% |
| Hepu Technology Development (Beijing) Co. Ltd. | 6 | 0 | 0 | 10 | 1 | 1 | 1 | 48% |
| JPMorgan Chase Bank, N.A. | 5 | 0 | 0 | 14 | 1 | 0 | 2 | 48% |
| Strong Force TX Portfolio 2018, LLC | 7 | 0 | 0 | 2 | 1 | 0 | 1 | 46% |
| LO3 Energy INC. | 6 | 1 | 0 | 4 | 1 | 2 | 3 | 42% |

As shown in the Table 3, Walmart Apollo, LLC and International Business Machines Corporation (IBM) have the highest number of patent applications. However, Walmart Apollo, LLC ranks first with a greater number of inventors, i.e., 36, an activity year of 2 years, and relative R&D capability of 100%. IBM has an activity year of 3 years and a research and development capability of 91%, ranking second. Furthermore, the number of patent application from the General Electric Company, Strong Force IoT Portfolio 2016, and Ethicon LLC are all seven, but the inventor numbers and the activity year of the General Electric Company are greater than the other two, ranking third. In summary, from the data analysis, it is known that the top three companies have certain authority in the research and development of energy blockchain technology.

Furthermore, Figure 6 shows the number of patent application for the top ten applicants in energy blockchain technology. With 2017 as the time limit, IBM and LO3 Energy started research and development on energy blockchain prior to 2017 and are the world's earliest companies to study energy blockchain, thus, they are very authoritative companies in the field of energy blockchain technology. After 2017, companies such as Walmart also started research on energy blockchain technology, and they had started to produce results. It is seen that the leaders in the field of energy blockchain technology are IBM and LO3, however, at present, many large companies have started to develop energy blockchain technologies which shows that it has good development prospects.

*4.4. IPC Analysis*

Before the IPC analysis, the technology classification of the energy blockchain addressed in the patent applications must be known. By using text mining on the abstract and a summary of the inventions of the found patents, the retrieved 319 patent applications are divided into several clusters. Figure 7 is a cluster map of the patent applications of energy blockchain technology retrieved from 319 patents in this study. The research and development in energy blockchain technology involves many aspects, including blockchain, data storage, electrical energy, energy consumption, computing device, communication module, and peer-to-peer, etc., as also shown in Figure 2. Among them, the number of

patent applications regarding blockchain is up to 80. It is also found that the number of patents in the combination of the system and method is 69. The number of patents related to data storage, energy power, and energy consumption is 47, 46, 44, respectively. The number of patents for computing devices, communication modules and peer-to-peer transactions are roughly around 20. Information from the cluster map of energy blockchain technology would help the analyst to understand the appearance of the IPC.

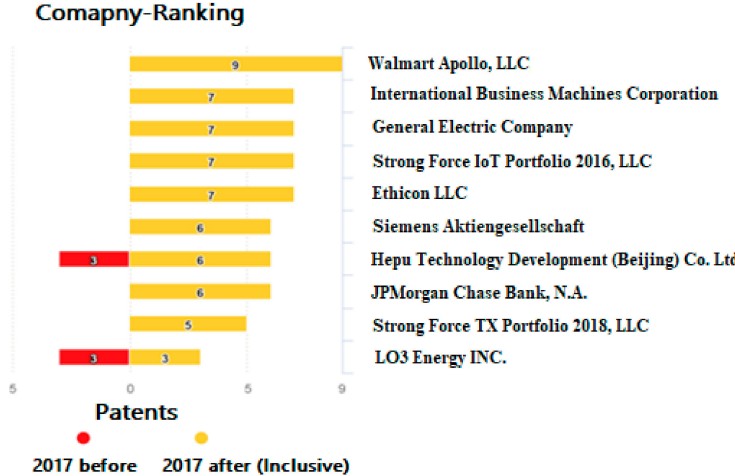

**Figure 6.** Number of patent application from the top ten applicants in energy blockchain technology.

Figure 8 shows the IPC chart of the patent applications in energy blockchain technology, and Table 4 describes the technical feature of the top five IPC patent applications in energy blockchain technology. Of the 319 related patent applications, a total of 39 main patents are classified as G06Q 50/06, that is, the number of patents in the technical field of electricity, gas, or water supply is the largest, occupying a high proportion. In addition, the technology categories with 12 and nine patent applications are G06Q 20/06 and H02J 3/38, respectively, which are private payment circuits, namely, involving electronic currency used only among participants of a common payment scheme and arrangements for parallel feeding a single network by two or more generators, converters, or transformers. The results show that the development of energy blockchain technology is mainly concentrated in these three fields. The development direction is consistent with the analysis of the cluster map for the patent applications, as shown in Figure 7. Therefore, these data have a guiding role on the research of energy blockchain technology for the companies and researchers.

Figure 9 shows the year variation of the top five IPC. The trend of each year based on the classification of the top five IPC technologies is further analyzed. Analysts use time points to observe the developmental trends of specific technologies and fully grasp important technical information. Only a small number of patents in this field began to be filed in 2016, indicating that energy blockchain technology has only begun to sprout from 2016. Although there were patent applications from 2016 to 2017, the number was only three, indicating that no breakthrough technology has been successfully developed in the field. The number of patent applications suddenly increased in 2018. Among them, G06Q 50/06 and G06Q 20/06 have the greatest number of patent applications. In addition, H02J 3/38 and H04L 9/32 have also increased after a year, which refer to the arrangements for parallel feeding a single network by two or more generators, converters, or transformers, and means for verifying the identity or authority of a user of the system, respectively. In summary, energy blockchain technology, so far, has focused on the research of G06Q 50/06, meaning setting and management of electricity, gas, or water supply.

The IPC number of patent applications refers to the number filed or owned by a company or country in a specific field or population ratio. Thus, by analyzing the IPC number of patent application of the top three applicants that have locked the energy blockchain technology based on the technical

characteristics of the IPC, it clearly shows the R&D focus and main patent layout direction of each company. Moreover, it is estimated that if there are a large number of patents, there are more innovative technologies, and therefore more competition. Table 5 shows the IPC number of patent applications of the top three applicants. The top three applicants have invested more in areas that are almost focused on the G06Q 50/06, namely, the setting and management of energy systems including electricity, gas, or water supply. It is also seen that other aspects of technical research are still being worked on. Using blockchain technology to solve the problem of Energy Internet will be the focus of research in the future.

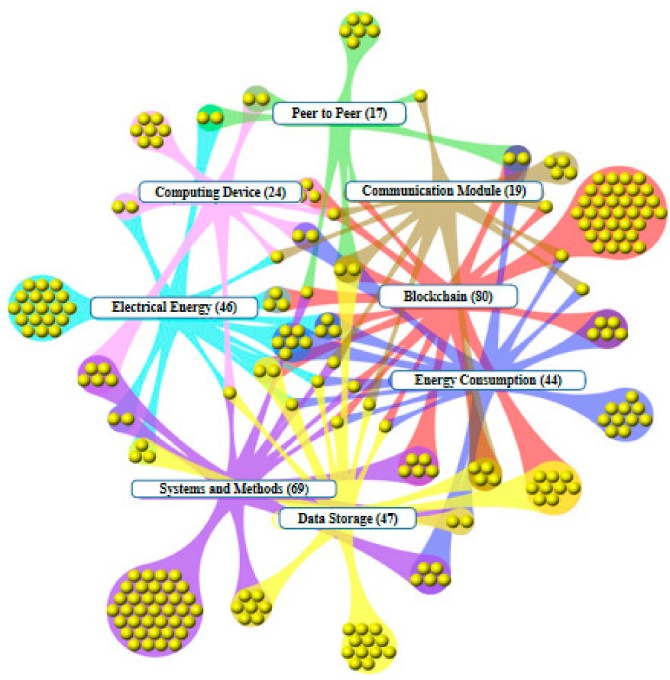

**Figure 7.** Cluster map of the patent applications in energy blockchain technology.

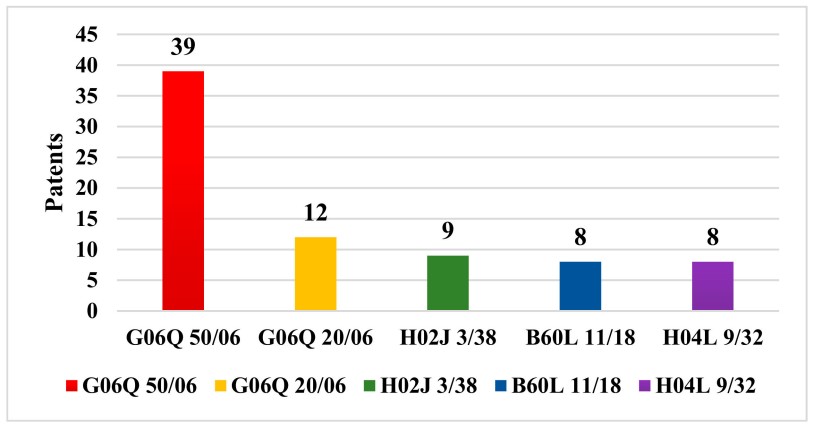

**Figure 8.** International patent classification (IPC) chart of the patent applications in energy blockchain technology.

**Table 4.** Technical feature of the top five International patent classification (IPC) patent applications in energy blockchain technology.

| IPC | Technical Feature | Numbers | The Relationship of Energy Blockchain Technology |
|---|---|---|---|
| G06Q 50/06 | Electricity, gas or water supply | 39 | Energy supply/ Electrical energy |
| G06Q 20/06 | Private payment circuits, e.g., involving electronic currency used only among participants of a common payment scheme | 12 | Blockchain/ Peer to peer |
| H02J 3/38 | Arrangements for parallel feeding a single network by two or more generators, converters, or transformers | 9 | Energy machine/ Computing device |
| B60L 11/18 | Using power supplied from primary cells, secondary cells, or fuel cells | 8 | Energy supply/Energy consumption |
| H04L 9/32 | Including means for verifying the identity or authority of a user of the system | 8 | Blockchain/ Communication module/ Security |

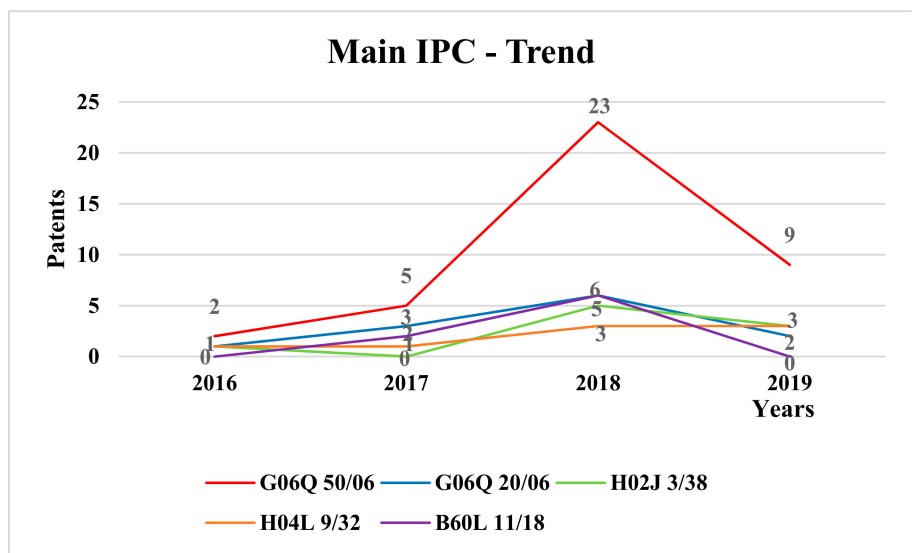

**Figure 9.** The year variation of the top five IPCs.

**Table 5.** The IPC number of patent applications of the top three applicants.

| Applicants\IPC | G06Q 10/06 | G06Q 20/06 | G06Q 20/38 | G06Q 50/06 | H02J 3/00 |
|---|---|---|---|---|---|
| Walmart Apollo, LLC | 0 | 1 | 0 | 2 | 0 |
| International Business Machines Corporation | 1 | 1 | 1 | 4 | 2 |
| General Electric Company | 1 | 0 | 1 | 1 | 0 |

*4.5. Analysis of Life Cycle*

From the number of patent applications and the number of applicants, it can be observed that the technology in this field is (1) the introduction period, (2) the growth period, (3) the mature period, or (4) the decline period [30–32].

(1) During the introduction period, the research and development unit discovered the key technology of the relevant topic. At this time, the development and research of the key technology were mastered by a few companies. Therefore, the number of related patent applications and patent applicants was not prominent, and there was a high concentration.

(2) During the growth period, various manufacturers and research units found the importance of this key technology. The key technology began to develop rapidly in the vertical and horizontal

directions. Therefore, the market gradually expanded, causing in the number of patent applications for this technology to skyrocket.

(3) During the maturity period, this key technology gradually approaches maturity. Most manufacturers and research units also invest considerable R&D manpower, and the related market development gradually becomes saturated. At this time, the rate of patent application growth is gradually increasing. Slowly, the number of patent applicants gradually stabilizes and is no longer rising.

(4) During the recession period, this key technology is not only fully saturated, but the related side branch technologies have also been deployed. At this time, the number of patent applicants has begun to decrease, the technology development is in a recession trend, and there is no longer the possibility of growth.

Therefore, analysis of the life cycle assesses whether the technology still has investment value, helps understand whether the key technology has been eliminated by the market, and helps determine whether any further costs need to be put into the technology.

Figure 10 shows the life cycle of energy blockchain technology. It is clearly seen that the number of patent applications started in 2015 and maintained steady growth from 2016 to 2017, which means that research units or manufacturers gradually started to notice business opportunities in this field. In 2018, the number of patent applications increased sharply, since the application of blockchain technology in the financial field allowed many R&D units to see the value of the blockchain and its advantages in the energy field. Quite a number of research units and manufacturers valued the research on energy blockchain technology and invested considerable R&D costs to expand their markets. In 2019, the number of patent applications decreased, and the reasons for this are considered to include: (1) The time of publication, many patent applications have not yet been published; and (2) the complexity of the energy field, related technologies have not yet satisfied the market. However, the advantages of the blockchain in the energy field are self-evident. It is known that most projects are in the early development stages, and research on key areas is still ongoing to improve the required scalability, decentralization, and security. Thus, it is believed that energy blockchain technology is still during the growth period, and in the next three to five years, there is still considerable room for development of patents in the later period.

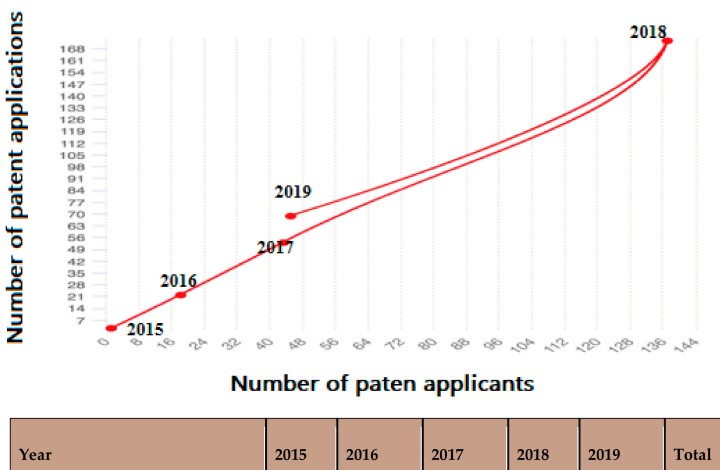

| Year | 2015 | 2016 | 2017 | 2018 | 2019 | Total |
|---|---|---|---|---|---|---|
| Number of patent applications | 2 | 22 | 53 | 173 | 69 | 319 |
| Number of patent applicants | 1 | 18 | 43 | 137 | 45 | 244 |

**Figure 10.** The life cycle of energy blockchain technology.

## 5. Conclusions and Recommendations

Blockchain is not just a technology, it is more a change of concept and a remodeling of some industries. The energy system is facing many challenges such as the popularization of renewable energy, carbon reduction pressure, and the development of a combined energy production and sales market, which has prompted energy companies and energy service providers to start their digital transformation. The application of blockchain in the decentralized energy trading has continued to grow recently, and it is expected that it could become a key technology to overturn the traditional centralized power market trading model. For the patent analysis, in this study, the research scope is limited to the patents applied to the setting, management, and trading of the energy system combined with the blockchain technology. The results of the patent analysis in energy blockchain technology have the following conclusions:

1. The number of the found patents used to be analyzed is 319, and among them, 24 patents were approved and 295 were public and disclosed.
2. The number of filed patent applications from 2017 to 2019 has increased significantly, and in 2018, the number of filed patent applications in the field of energy blockchain technology reached 173.
3. United States (USA) has the highest share of energy blockchain technology patent applications, up to 59%, followed by Germany (DE) with 27.9%, and China (CN) ranks third, with the share of 6%, Great Britain (GB) and Israel (IT) have 5% and 2% respectively,
4. The participants are not from traditional specific fields but include electric power manufacturers, computer software companies, e-commerce companies, and even many new companies devoted to blockchain technology.
5. Walmart Apollo, LLC has a greater number of inventors of 36, an activity year of 2 years and relative R&D capability of 100%, ranking first; and IBM has an activity year of 3 years and has a research and development capability of 91%, ranking second.
6. IBM and LO3 energy started earlier in this field, and their patent output is also more prominent.
7. The IPC is mainly concentrated in G06Q 50/06, having a total of 39 main patents, which belong to the technical field of the setting and management of the energy system including electricity, gas, or water supply.
8. According to the life cycle analysis, energy blockchain technology is still during the growth period, and there is still considerable room for development of patents in the later period.

From the entire analysis process, the application of blockchain technology with the energy field can bring greater flexibility and better performance to the energy trading market. Currently, most projects are in the early development stages, and research on key technology areas is still ongoing, which will allow the required scalability, decentralization, and security. With respect to the challenges of energy blockchain technology discussed in many references and cooperate with our results of the patent analysis, the recommendations of patent layout strategy for the energy blockchain technology are provided as follows:

1. Different types of blockchain, such as the consortium blockchain and the private blockchain, can also combine their own advantages to improve the efficiency of the energy transaction process.
2. In the energy blockchain system, a better incentive mechanism is needed to encourage the users to join the energy blockchain system and encourage the renewable energy generation to bring benefits to users.

Although most of the current projects are small pilot projects, considerable results have been achieved. Blockchain technology will bring revolutionary changes to the energy market and will face a variety of challenges in achieving market penetration, including legal, regulatory, and competitive obstacles. More research programs, trials, projects, and collaborations will show whether the technology can realize its full potential, prove its commercial viability, and eventually be adopted by the mainstream.

The management mechanism of the current energy blockchain technology is an urgent issue that needs to be resolved. In general, energy blockchain technology is the key to the energy Internet. We also have reasons to believe that in the future, energy blockchain technology will continue to grow into a global model.

At present, many start-up companies have cooperated with energy companies to invest in the development and demonstration of blockchain energy platforms. Innovative technologies in recent years such as big data, the Internet of Things (IoT), artificial intelligence (AI), and other emerging technologies are also accelerating the development and industrial innovation of energy blockchain technology. As an example of taking balancing energy supply and demand, after collecting a large amount of electricity and power generation data through various IoT sensing devices, the AI technology can accurately forecast demand and the amount of renewable energy power generation, which will make the operation of the energy system more predictable and reliable.

The above results of the patent analysis are useful for policy implications on the patent distribution in energy blockchain technology. First, as the energy system becomes more complex and decentralized, the technical requirements for big data analysis and Internet of Things (IoT) are also increasing [43]. The directions of patent application of the energy blockchain technology include the following: optimizing asset management, improving operating efficiency, balancing supply and demand, increasing the flexibility of the power grid, increasing the capacity of renewable energy, and developing innovative services. Second, with the evolution of the Energy Internet, the energy industry is changing. In the future, there will be more microgrids, virtual power generation, and decentralized renewable energy using blockchains. Thus, the directions of patent applications in energy blockchain technology can be applied to electric vehicles, smart houses, and smart communities. Third, it is found that most of the leading applicants are from companies. Considering the large research ability of the universities and research institutes, it is necessary to strengthen the construction of an industry/university/research (IUR) platform, thus providing more talent in related fields [43].

This study makes an important contribution towards identifying development trends and investment opportunities using patent analysis that has not appeared in existing methodologies of literature review. Through this search, analysis, and interpretation of patents in energy blockchain technology, further research direction on the technical curve and context of energy blockchain technology based on patent analysis can be conducted. With the discussions, we strive to grasp the research priorities of various countries in energy blockchain technology to provide further comprehensive references for enterprises and scholars.

Despite interesting findings and contribution, there are still some limitations as follows: (1) When searching related patents, certain limitations are appeared due to the limitation of time and geographical patent terms. (2) The number of patents within the past year could be incomplete due to the 18 month lag period of patents. (3) A patent application does not necessarily result in the approved patent through the examination. However, because the number of the approved patents is still small, we analyze the number of patent applications by using the method analyzing the number of the approved patents in the past. Therefore, there is still an error in the evaluation of the relative R&D capability, but it has little effect on the analysis of development trends. (4) Due to rapid developments in energy blockchain technology, the number of patent applications and approved patents have also increased rapidly, and the development trend in this field is expected to change dynamically with the advancement of time.

In future research, we plan to explore more in-depth insights into the dynamic variations of development trends through patent analysis. The above limitations serve as the direction for further in-depth research. We can conduct case investigations of patent documents in leading institutions, and perhaps interview relevant patent technical experts to obtain more advanced answers in the layout direction.

**Author Contributions:** L.-Y.H. and M.-H.W contributed the research concept; L.-Y.H. and J.-F.C. collected the data; L.-Y.H. and J.-F.C. analyzed the data and wrote the paper; L.-Y.H. revised the paper; T.-C.L. and M.-H.W. provided the advised the research. All authors have read and agreed to the published version of the manuscript.

**Funding:** This research received funding from the Putian university talent introduction [2019108, 2019003], support by the Education and Scientific Research Project for the young teachers in the Department of Education of Fujian province [No. JAT190576]; Natural Science Foundation of China (No. 61741111); Program for New Century Excellent Talents in Fujian Province University (Tao Yan); In part by Natural Science Foundation of Fujian (No. 2019J01816) and Natural Science Foundation of Jiangxi (20181BAB202011).

**Conflicts of Interest:** The authors declare no conflict of interest.

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
