# Peer review of "A Study on the Development Trends of the Energy System with Blockchain Technology Using Patent Analysis"

_sustainability, doi:10.3390/su12052005_

Round 1

Reviewer 1 Report

Dear Authors,

I have read your work and based on that my suggestions are as follows:

Introduction section need to be more informative. In literature review more discussion needed for 2.3 section as this is the heart of the paper Figure 3...avoid red color fonts. Table no. 2 equations are not in standard format Figure 6 legends are not clear

Good luck!!!

Author Response

The comments from the reviewer are very constructive to our paper. The paper is revised in accordance with the reviewer’s comments and has marked the corrections using yellow color in the revised word file. The attached are the revision and the reply letter for the comments. There are listed as follows:

Reviewer 1

Q1. Introduction section need to be more informative.

Answer:

Thank you and this comment is well received. We have added more informative description in introduction section 1 as shown in revised version with yellow mark.

Q2. In literature review more discussion needed for 2.3 section as this is the heart of the paper.

Answer:

Thank you and this comment is well received. We have added more informative description in 2.3. patent analysis of literature review section 2 as shown in revised version with yellow mark.

Q3. Figure 3...avoid red color fonts.

Answer:

Thank you and this comment is well received. We have replaced Figure 3 as new one.

Q4. Table no. 2 equations are not in standard format.

Answer:

Thank you and this comment is well received. We have modified equations as standard format.

Q5. Figure 6 legends are not clear.

Answer:

Thank you and this comment is well received. We have modified Figure 6 as clearly one.

Reviewer 2 Report

The present paper uses patent analysis technology to explore the application of blockchain in the setting, management and trading of the energy system, and making a series of the analyzed charts to reveal the development trend of the energy system with blockchain technology. The topic presented in this work is really interesting. However several challenges are required:

I analyze the single sections:

Abstract has inappropriate structure. I suggest to answer the following aspects: - general context - novelty of the work - methodology used (describe briefly the main methods or treatments applied) - main results and related interpretations.

Introduction: This section should briefly place the study in a wide context and emphasize why it is relevant carrying out the analysis. It should define the purpose of the work and its significance. In this perspective, this section is too succinct and fails to effectively point out the relevance of your contribution towards the existing literature. I would see much general emphasis on sustainability and financial economics (e.g. renewable energy). Some literature to start with:

Some references to look at are:

Enabling investment for the transition to a low carbon economy: government policy to finance early stage green innovation

Financing renewable energy: Who is financing what and why it matters 

Green investment strategies and bank-firm relationship: a firm-level analysis

Methodology is unclear. Initially a short resume can be proposed to explain several steps. The methodology used must be linked to the existing literature. what is its potential? its limit? Results are not always linked to the methodology. Please define the relationship and relate your finding with the relevant literature.

Conclusion are extremely succinct. Please provide policy implications for your study.

Author Response

The comments from the reviewer are very constructive to our paper. The paper is revised in accordance with the reviewer’s comments and has marked the corrections using yellow color in the revised word file. The attached are the revision and the reply letter for the comments. There are listed as follows:

Reviewer 2

Q1.Abstract has inappropriate structure. I suggest to answer the following aspects: - general context - novelty of the work - methodology used (describe briefly the main methods or treatments applied) - main results and related interpretations.

Answer:

  Thank you and this comment is well received. We have re-wrote the Abstract section as shown in revised version with yellow mark according to the reviewer’s suggestions.

Q2.Introduction: This section should briefly place the study in a wide context and emphasize why it is relevant carrying out the analysis. It should define the purpose of the work and its significance. In this perspective, this section is too succinct and fails to effectively point out the relevance of your contribution towards the existing literature. I would see much general emphasis on sustainability and financial economics (e.g. renewable energy). Some literature to start with:

Some references to look at are:

Enabling investment for the transition to a low carbon economy: government policy to finance early stage green innovation

Financing renewable energy: Who is financing what and why it matters 

Green investment strategies and bank-firm relationship: a firm-level analysis

Answer:

  Thank you and this comment is well received. We have re-wrote the introduction section in introduction section 1 as shown in revised version with yellow mark according to the reviewer’s suggestions. Namely, we have defined the purpose of this work and discussed its significance, and also pointed out the relevance of our contribution towards the existing literature. Moreover, we have added some references about general emphasis on sustainability and financial economics (e.g. renewable energy) in the references [4-6].

  1. Owen, R.; Brennan, G.; Lyon, F. Enabling investment for the transition to a low carbon economy: government policy to finance early stage green innovation. Current Opnion in Environmental Sustainability, 2018, 31, 137-145.
  2. Mazzucato, M.; Semieniuk, G. Financing renewable energy: Who is financing what and why it matters, Technol. Forecast. Soc. Change, 2018, 127, 8-22.
  3. Falcone, P. M. Green investment strategies and bank-firm relationship: A firm-level analysis, Economics Bulletin, 2018, 38(4), 2225-2239.

Q3.Methodology is unclear. Initially a short resume can be proposed to explain several steps. The methodology used must be linked to the existing literature. what is its potential? its limit? Results are not always linked to the methodology. Please define the relationship and relate your finding with the relevant literature.

Answer:

  Thank you and this comment is well received. We have added a short resume to explain several steps, and added more descriptions in research methodology section 3.1 as shown in revised version with yellow mark according to the reviewer’s suggestions. Namely, we have linked the methodology to the existing literature, addressed its potential and its limit, and carefully related our finding with the relevant literature.

Q4. Conclusion are extremely succinct. Please provide policy implications for your study.

Answer:

  Thank you and this comment is well received. We have added more descriptions in conclusion section 5 as shown in revised version with yellow mark according to the reviewer’s suggestions. Namely, we have added more results as summary, provided a policy implications, and addressed the contribution and limitation of this study as well as the future work.

Round 2

Reviewer 2 Report

Dear authors,

I found your revised version of the manuscript really improved. Congratulations.

I want just point out a minor issue to make your implications adherent to the main literature on the topic. 

I would relate the policy implications with literature on energy transition.

Some examples: You stated: "Considering the large research ability of the universities and research institutes, it’s necessary to strengthen the construction of industry-university-research (IUR) platform, thus providing more talent in related fields" and could relate it with some results of the following work:  Policy mixes towards sustainability transition in the Italian biofuel sector: Dealing with alternative crisis scenarios

In the same vein, you said " ...as the energy system becomes more complex and decentralized" and could look at this interring article: Decentralized energy system for clean electricity access.

Looking forward to see your article published.

Author Response

Reviewer 2

I found your revised version of the manuscript really improved. Congratulations.

I want just point out a minor issue to make your implications adherent to the main literature on the topic. 

I would relate the policy implications with literature on energy transition.

Some examples: You stated: "Considering the large research ability of the universities and research institutes, it’s necessary to strengthen the construction of industry-university-research (IUR) platform, thus providing more talent in related fields" and could relate it with some results of the following work:  Policy mixes towards sustainability transition in the Italian biofuel sector: Dealing with alternative crisis scenarios

In the same vein, you said " ...as the energy system becomes more complex and decentralized" and could look at this interring article: Decentralized energy system for clean electricity access.

 Looking forward to see your article published.

Answer:

  Thank you very much for your kind encourage and suggestion. This comment is well received. We have added two references [43] [44] in right place of conclusion section 5 to relate the policy implications with literature on energy transition as shown in revised version with green mark accordingly.

  1. Alstone, P.; Gershenson, D.; Kammen, D. Decentralized energy systems for clean electricity access. Nature Clim Change2015, 5, 305–314.
  2. Falcone, M.; Lopolito, A.; Sica, E. Policy mixes towards sustainability transition in the Italian biofuel sector: Dealing with alternative crisis scenarios, Energy Research & Social Science, 2017, 33, DOI: 10.1016/j.erss.2017.09.007.
